# Association between NTRK2 Polymorphisms, Hippocampal Volumes and Treatment Resistance in Major Depressive Disorder

**DOI:** 10.3390/genes14112037

**Published:** 2023-11-03

**Authors:** Marco Paolini, Lidia Fortaner-Uyà, Cristina Lorenzi, Sara Spadini, Melania Maccario, Raffaella Zanardi, Cristina Colombo, Sara Poletti, Francesco Benedetti

**Affiliations:** 1Psychiatry and Clinical Psychobiology Unit, Division of Neuroscience, IRCCS San Raffaele Scientific Institute, 20132 Milan, Italy; 2Mood Disorders Unit, IRCCS San Raffaele Scientific Institute, 20132 Milan, Italy; 3Faculty of Medicine, Vita-Salute San Raffaele University, 20132 Milan, Italy; 4Faculty of Psychology, Vita-Salute San Raffaele University, 20132 Milan, Italy

**Keywords:** treatment-resistant depression, antidepressant efficacy, hippocampal volumes, BDNF, TrkB, NTRK2

## Abstract

Despite the increasing availability of antidepressant drugs, a high rate of patients with major depression (MDD) does not respond to pharmacological treatments. Brain-derived neurotrophic factor (BDNF)-tyrosine receptor kinase B (TrkB) signaling is thought to influence antidepressant efficacy and hippocampal volumes, robust predictors of treatment resistance. We therefore hypothesized the possible role of BDNF and neurotrophic receptor tyrosine kinase 2 (*NTRK2*)-related polymorphisms in affecting both hippocampal volumes and treatment resistance in MDD. A total of 121 MDD inpatients underwent 3T structural MRI scanning and blood sampling to obtain genotype information. General linear models and binary logistic regressions were employed to test the effect of genetic variations related to *BDNF* and *NTRK2* on bilateral hippocampal volumes and treatment resistance, respectively. Finally, the possible mediating role of hippocampal volumes on the relationship between genetic markers and treatment response was investigated. A significant association between one *NTRK2* polymorphism with hippocampal volumes and antidepressant response was found, with significant indirect effects. Our results highlight a possible mechanistic explanation of antidepressant action, possibly contributing to the understanding of MDD pathophysiology.

## 1. Introduction

Major depressive disorder (MDD) is a highly prevalent psychiatric condition, associated with relatively high rates of disability and mortality [1]; antidepressants, mainly selective serotonin reuptake inhibitors (SSRIs) and serotonin and norepinephrine reuptake inhibitors (SNRIs), are currently used as the first-line pharmacological treatment of MDD [2]. However, response and remission rates to first-line treatments are modest [3], and antidepressant switch, treatment augmentation strategies or even electroconvulsive therapy are often required [4].

In recent years, it has become clear that numerous drugs have antidepressant properties without directly acting on monoamine reuptake. Additional putative antidepressant mechanisms of action include direct modulation of serotoninergic [5,6] and noradrenergic receptors [7], action on glutamatergic N-methyl-D-aspartate (NMDA) receptors [8,9] and modulation of γ-aminobutyric acid (GABA) signaling [10,11]. Furthermore, novel antidepressants acting on opioid, trace amines, cannabinoid, α-amino-3-hydroxy-5-methyl-4-isoxazolepropionic acid (AMPA) and metabotropic glutamatergic receptors, as well as immuno-modulatory drugs for depression, are currently being developed [12,13].

This plethora of putative antidepressant mechanisms highlights the limitation of the old monoamine hypothesis of depression—that is, that depression stems from alterations of monoaminergic neurotransmission [14]—and suggests the existence of a common final pathway through which different pharmacological mechanisms lead to antidepressant effects.

A leading candidate for the role of a common pathway of antidepressant action is the neuroplasticity hypothesis [15]: this theory postulates a crucial role for brain-derived neurotrophic factor (BDNF) and its receptor, tyrosine receptor kinase B (TrkB), in both MDD pathophysiology and antidepressant efficacy [16]. BDNF is a member of the nerve growth factor family, which is crucial during brain development but is also expressed in adulthood, where it regulates synaptic plasticity, neuronal function and survival [17]. Lower levels of BDNF protein and mRNA have been identified in postmortem brain tissue of depressed patients, particularly in the hippocampus [18,19], as well as of people who died of suicide [20]; increased *BDNF* gene metilation and reduced BDNF expression has been found in peripheral blood mononuclear cells of depressed and suicidal patients [21]. Likewise, reduced levels of BDNF receptor TrkB, of its activated phosphorylated form and mRNA, have been identified in postmortem brains of depressed individuals [20,22,23]. Further, alterations of BDNF-TrkB signaling in different brain regions including the hippocampus have been shown to mediate the relation between lipopolysaccharide-induced inflammation and depression-like phenotypes in animal models.

While reductions of BDNF are not specific to depression, having been reported also in other psychiatric or neurological conditions, the BDNF-TrkB pathway seems to be crucially involved in antidepressant response [16]. Monoaminergic antidepressants, ketamine and electroconvulsive therapy (ECT) all induce increases in brain BDNF, both in mouse models and in humans [24,25,26], and BDNF signaling through TrkB appears to be crucial for antidepressant effects [27]. Deletion of both BDNF or TrkB in hippocampal neurons inhibits the effect of antidepressants in mouse models [28,29]; at the same time, only deletion of TrkB and not of BDNF in the dorsal raphe reduces antidepressant responses [30]. According to a recent suggestive hypothesis, different classes of antidepressants might exert, at least partially, their activity directly binding to the TrkB receptor, stabilizing its trans-membrane configuration and promoting BDNF-mediated signaling [16,31]; both typical monoaminergic antidepressants (selective serotonin reuptake inhibitors (SSRIs) and serotonin and norepinephrine reuptake inhibitors (SNRIs)) and ketamine have been found to have substantial binding affinity for TrkB [31].

Reduced hippocampal volumes are among the few robust structural MRI predictors of poor antidepressant response and treatment resistance [32,33]. Reduced hippocampal volumes in MDD may reflect impaired neuroplasticity and neurogenesis [14], thought to be crucial in antidepressant activity [34]; indeed, damage to hippocampal neurogenesis via X-irradiation of the mouse brain inhibits the effects of antidepressants [35]. BDNF, one of the main neurotrophic factors in the adult brain [17,36], plays a crucial role in hippocampal neurogenesis, and its levels have been repeatedly associated with hippocampal volumes [37,38,39], albeit somewhat inconsistently [40]. While the effect of *BDNF* polymorphisms on hippocampal volumes has been the subject of extensive research [41,42,43], the effect of genetic variation in the region coding for its receptor has been much less investigated; one study, however, reported a significant association in a general population sample [44].

The effects of *BDNF* genetic polymorphisms on antidepressant efficacy have been widely investigated [45]. One particular polymorphism, Val66Met, received much attention as it influences BDNF trafficking and release and affects antidepressant response in mouse models [46]; however, such effects do not translate so clearly to humans, as heterozygous subjects for Val66Met appear to have a better response to antidepressants than either Val or Met homozygous subjects [16,47].

Effects of genetic variation in the TrkB gene—*NTRK2*—on the other hand, have been much less studied. Neurotrophic receptor tyrosine kinase 2 (*NTRK2)* polymorphisms have been shown to influence response to mood-stabilizing agents in bipolar patients [48]. Concerning antidepressant response, while some studies fail to identify an association with polymorphisms mapping to the *NTRK2* gene [49,50], others do report such an association [51,52]. T/T carriers in the rs1565445 *NTRK2* polymorphism were reported to have higher risk of developing treatment-resistant depression compared to patients with genotype C/C and T/C [51]. Hennings and colleagues found nine *NTRK2* single nucleotide polymorphisms (SNPs) to be associated with antidepressant response patterns (response at 5 weeks and remission at discharge), and for three of such SNPs, replicated the association in independent samples [52]. Polymorphisms in the *NTRK2* region have also been associated with antidepressant treatment-emergent suicidal ideation, an indirect marker of poor treatment efficacy [53].

Here, our aim was to investigate the interplay between polymorphism mapping of the *BDNF* and *NTRK2* gene region, hippocampal volumes and treatment resistance in a sample of MDD inpatients in a real-world clinical setting. We hypothesized that genetic variants of *BDNF* or *NTRK2* genes would associate with hippocampal volume which, in turn, would affect treatment resistance.

## 2. Materials and Methods

### 2.1. Participants

Our study was performed on a sample of 126 inpatients with a diagnosis of MDD suffering from a major depressive episode without psychotic features (DSM-5 criteria) admitted to the Mood Disorders ward of San Raffaele Hospital in Milan. Patients had been referred for hospital specialized clinical treatment of depression by their general practitioners or psychiatrists in charge. Treatment was administered by staff psychiatrists upon clinical need. Prescribed pharmacotherapies can be seen in Appendix A.

Exclusion criteria were current diagnosis of any additional psychiatric disorder, including alcohol and/or substance dependence or abuse in the last 6 months, intellectual disability, pregnancy, major medical and neurological disorders.

Treatment resistance status was obtained for each patient from the clinical charts. Patients were defined as being treatment-resistant if, during the current depressive episode, they had failed to respond to at least two separate antidepressant treatments with different mechanisms of action, administered to a dose equal or superior to the minimum licensed dose for at least 4 weeks [54,55].

After a complete description of the study was given to the participants, written informed consent was obtained. All the research activities were approved by the local ethical committee. The described work has been carried out in accordance with The Code of Ethics of the World Medical Association (Declaration of Helsinki) for experiments involving humans.

### 2.2. Genetic Analysis

All patients underwent venous blood sampling to obtain genetic data. Blood samples were genotyped using the Infinium PsychArray 24 BeadChip (Illumina, Inc., San Diego, CA, USA), which is a cost-effective, high-density microarray developed in collaboration with the Psychiatric Genomics Consortium and containing 595,427 genetic markers associated with psychiatric disorders (https://www.illumina.com/products/by-type/microarray-kits/infinium-psycharray.html, accessed on 30 September 2023). Specifically looking at genetic variants mapping to the *BDNF* or *NTRK2* genes regions, 46 markers for *NTRK2* and 34 for *BDNF* are included in this array. Genotype quality control was conducted with PLINK1.9 [56]. First, markers with minor allele frequency (MAF) <5%, call rate <95% or deviant from Hardy–Weinberg equilibrium at *p* < 10^−6^ were excluded, leaving 31 markers for *NTRK2* and 17 for *BDNF*.

Secondly, individuals with discrepant genotyped sex information, genotype rate <95% or outlying autosomal heterozygosity (Fhet > ±0.2) were removed. Finally, relatedness of participants was checked by excluding individuals with a degree of recent shared ancestry (i.e., identity by descendent, IBD) >0.1875, which is the threshold corresponding to halfway between third- and second-degree relatives [57]. Therefore, the final sample consisted of 121 MDD patients.

### 2.3. Brain Imaging

T1-weighted images were acquired on two 3.0 Tesla scanners: 44 patients underwent a 3T MRI scan in a Gyroscan Intera scanner, Philips, The Netherlands, employing an 8-channel SENSE head coil (T1-weighted MPRAGE sequences: TR 25.00 ms, TE 4.6 ms, field of view FOV = 230 mm, matrix = 256 × 256, in-plane resolution 0.9 × 0.9 mm, yielding 220 transversal slices with a thickness of 0.8 mm); 77 patients were scanned in a 3T Ingenia CX scanner, Philips, The Netherlands, using a 32-channel sensitivity-encoding SENSE head coil (T1-weighted MPRAGE sequence: TR 8.00 ms, TE 3.7 ms, field of view FOV = 256 mm, matrix = 256 × 256, in-plane resolution 1 × 1 mm, yielding 182 transversal slices with a thickness of 1 mm).

Images were visually inspected as part of the quality check procedure, and then processed using the Computational Anatomy Toolbox (CAT12) preprocessing pipeline [58] for SPM12 (https://www.fil.ion.ucl.ac.uk/spm/software/spm12/, accessed on 30 September 2023) in Matlab R2016b, which also allows for the extraction of ROIs tissue volumes. This included segmentation into gray matter, white matter and cerebrospinal fluid, bias regularization, non-linear modulation and normalization to MNI space using DARTEL to a 1.5 mm isotropic MNI template. Bilateral hippocampal volumes were estimated according to the Neuromorphometric Atlas and converted into percentage of total intracranial volume (TIV) with the formula: (Hippocampal Volume × 100)/TIV [59]. We will refer to the result of this calculation as normalized hippocampal volume.

### 2.4. Statistical Analysis

General linear models and binary logistic regression analyses were performed with StatSoft Statistica 12 (Tulsa, OK, USA); parameter estimates were obtained with least squares maximum likelihood procedures. The significance of the effects was calculated with the likelihood ratio statistic. For mediation analyses, we used Hayes’ SPSS “Process” Macro, version 4.0 with IBM SPSS statistics for Windows, Version 26.0 (IBM SPSS Statistics for Windows, Version 26.0., Armonk, NY, USA: IBM Corp.) [60].

The effect of genetic variations in *BDNF* and *NTRK2* on bilateral hippocampal volumes were tested in the context of general linear models, entering left or right hippocampal volume as dependent variables, polymorphisms as independent variables and age, sex and MRI scan as nuisance covariates. The effect of each polymorphism was tested in a dominant (MM versus Mm + mm), recessive (MM + Mm versus mm) and over-dominant genetic model (MM + mm versus Mm) [61]. When no homozygous subject for the minor allele was identified, only the comparison between the remaining two groups was performed. This resulted in 272 comparisons, setting the Bonferroni threshold for statistical significance to *p* < 0.00018 (Appendix A) [62].

The effect of genetic polymorphisms on treatment resistance status was tested with binary logistic regressions, entering treatment resistance as a dichotomous dependent variable, polymorphisms as independent variables and age and sex as nuisance covariates. Again, the dominant, recessive and over-dominant effects of polymorphisms were tested, without probing the effect of genetic models with fewer than three subjects in one group; this resulted in 128 tests, with a Bonferroni-corrected threshold of significance of *p* < 0.00039 (Appendix A).

The effect of hippocampal volumes on treatment resistance status was again tested in the context of logistic regression, with resistance as a dependent variable, left or right hippocampal volumes as independent variables and age, sex and MRI scan as nuisance covariates.

After reviewing the results of the previous analyses, we tested the indirect effect of significant polymorphisms on treatment resistance through hippocampal volumes in a mediation model, entering age, sex and MRI scan as nuisance covariates. Given the high correlation between left and right hippocampal volumes (r = 0.898), a possible indirect effect of left hippocampal volume was also tested.

## 3. Results

Clinical and demographic characteristics of the sample can be seen in Table 1. Patients with treatment-resistant depression had significantly longer hospital stays and lower bilateral hippocampal volumes. Polymorphism distribution across subjects is presented in Appendix A.

After applying Bonferroni HWE correction for multiple comparisons, one *NTRK2* polymorphism (rs1948308) was found to be significantly associated with right hippocampal volume in the over-dominant genetic model; heterozygote patients had significantly lower right hippocampal volume compared to homozygotes (*p* = 0.000082).

Testing the effect of all polymorphisms on treatment resistance, rs1948308 was again found to be significantly associated with treatment resistance in the over-dominant genetic model; heterozygote patients for the polymorphism had a significantly higher chance of being classified as treatment-resistant than AA or GG homozygotes (*p* = 0.000075).

Bilateral hippocampal volumes were significantly associated with treatment resistance (L: *p* = 0.000149; R: *p* = 0.000420).

The mediation model confirmed that rs1948308 affected bilateral hippocampal volumes (R: *p* < 0.001; L: *p* = 0.002), which in turn influenced treatment resistance status (R: *p* = 0.024; L: *p* = 0.006). Therefore, a significant indirect effect of rs1948308 heterozygosis on treatment resistance through bilateral hippocampal volumes was identified (R: 95% BCa CI [0.060, 0.853] (Figure 1); L: 95% BCa CI [0.079, 0.800]); in both cases, the direct path of the model also remained significant (R: *p* = 0.0031; L: *p* = 0.0018) (Figure 1).

No statistically significant results were identified for polymorphisms involving the *BDNF* gene region (see Appendix A).

## 4. Discussion

The main finding of the present study is the association between one polymorphism mapping to the *NTRK2* region and antidepressant treatment resistance, mediated by hippocampal volumes. Reduced hippocampal volumes are a robust finding in neuroimaging studies of MDD patients [63] and are thought to reflect impaired neuroplasticity and neurogenesis processes [14], both of which are strongly affected by the BDNF-TrkB signaling pathway [64]. BDNF is a highly expressed neurotrophic factor in the adult brain, and through its high-affinity receptor—TrkB—plays a crucial role in regulating neuronal survival and synaptic plasticity [65]; furthermore, BDNF-TrkB have also been shown to play a pivotal role in promoting neurogenesis, the formation of new neurons that occurs in some regions of the adult brain, including the hippocampus [64]. Chronic stress reduces hippocampal survival and proliferation of progenitor cells, as well as dendritic complexity [66], possibly through its effect on BDNF [67]. Indeed, a model of depressive symptomatology has been proposed, postulating a stress-induced reduction in BDNF signaling, reflected in lower hippocampal volumes, and restored through successful antidepressant therapy [68].

Accordingly, genetic variations both in *BDNF* and in *NTRK2* have been associated with hippocampal volumes, in healthy and psychiatric samples alike [42,43,44]. Lower bilateral hippocampal volumes, in turn, are one of the best replicated markers of poor antidepressant efficacy and of treatment resistance [32]. Our data suggest that *NTRK2* genetic variants affect antidepressant resistance at least partially through their effect on hippocampal volumes.

In recent years, it has become apparent that immune/inflammatory alterations are involved in MDD pathophysiology [69,70] and depressive symptomatology is now thought to be a consequence of low-grade chronic inflammatory status, at least in a subset of patients [71]. Inflammation has been suggested to lead to depressive symptomatology through its effect on brain integrity including a reduction in hippocampal volumes [72,73], and has also been linked to poor antidepressant efficacy [74]. Recently, it has been shown that lipopolysaccharide-induced inflammation alters BDNF signaling in widespread brain regions including the hippocampus [75]; the BDNF-TrkB pathway, therefore, might represent the link between alterations in immune/inflammatory status and hippocampal volume, a putative neurogenesis/neuroplasticity biomarker [37,38,39] robustly linked to depressive symptomatology and treatment efficacy [32].

It has been suggested that BDNF high-affinity receptor, TrkB, encoded by the *NTRK2* gene [76], plays a crucial role in biological processes leading to antidepressant response [77]. *NTRK2*’s rare esonic mutations impair hippocampal synaptogenesis and associate with neurobehavioral abnormalities [78]. Recently, various antidepressant compounds of different classes (including SSRIs, SNRIs and ketamine) have been shown to be able to directly bind TrkB, enhancing its signal transduction [31]; a provocative hypothesis, therefore, has been put forward, postulating a direct contribution of TrkB binding to the antidepressant (AD) mechanism of action [79]. A more conservative hypothesis postulates that AD-induced increase in monoaminergic neurotransmission may enhance brain neuroplasticity and neurogenesis, thus leading to antidepressant effects; this framework also provides an explanation for the latency of action of “classic” monoaminergic antidepressants [80]. The BDNF-TrkB signaling pathway is commonly regarded as a crucial mediator of this process [15], and growing evidence also links it to the action of fast-acting AD treatments, such as ECT and ketamine [9,81]. Regardless of the precise pathways through which ADs with various mechanism of actions exert their clinical effect—whether through a direct interaction with the TrkB receptor or an indirect increase in brain BDNF levels—BDNF-TrkB signaling emerges as a crucial molecular pathway of antidepressant action.

The polymorphism we identified was in a non-coding region, suggesting its effect on hippocampal volumes and treatment resistance are linked to genetic regulatory processes [82]. Interestingly, rs1948308, in addition to having been identified in pharmacogenetic studies of opioid use disorder [83], had previously been linked to antidepressant efficacy, although it failed to survive multiple comparison correction [52]. rs1948308 has also been robustly associated with several psychiatric phenotypes, such as neuroticism and mood swings [84,85].

We observed lower hippocampal volumes and higher rates of treatment resistance in heterozygous subjects compared to homozygous for either allele (over-dominant genetic model) [61]. Although this finding might appear counter-intuitive and certainly warrants further investigation, studies investigating the effect of *BDNF* Val66Met polymorphism (known to affect intracellular BDNF trafficking and release) on antidepressant efficacy also report significant effect in heterozygous subjects, with Val/Met carriers characterized by better treatment response [16,47]. However, no polymorphism in the *BDNF* region, including Val66Met, affected hippocampal volumes or treatment resistance in our sample. Further, although not surviving correction for multiple comparisons, we observed a general effect of the recessive allele on both TRD and hippocampal volumes (Appendix A). This may suggest a possible effect of the recessive allele that may not be visible here because of the reduced sample size but that could have a more easily interpretable biological meaning. To better understand this issue, in vitro studies could help highlight the biological effect of this genetic variation.

The observation that we found a significant result for TrkB and not BDNF genetic variants may suggest that the association between hippocampal volume and treatment resistance is not associated with the amount of BDNF released, influenced among others by the Val66Met polymorphism, but instead to its link with its receptor. TrkB is a tyrosine kinase receptor that, upon binding to its ligands—mainly BDNF but also neurotrophin 3 and 4—undergoes dimerization and starts signal transduction [86]; it is widely expressed in both peripheral and central nervous systems, including the hippocampus [87], and regulates cell growth and survival [85]. Its expression has been observed, however, in other brain cell types, such as astrocytes and microglia, with less clear biological functions [88,89]. Interestingly, a truncated form of the receptor, without the catalytic tyrosine kinase domain, is also expressed in the brain, and appears to have numerous regulatory functions [90]. Furthermore, TrkB expression on cell surface is highly regulated via numerous modulatory mechanisms, such as cAMP levels, N-glycosylation and receptor transactivation, and this strongly influences cell response to BDNF, even in short time periods [91]. All of these complex and highly integrated regulatory mechanisms of the BDNF-TrkB signaling, both on neurons and on other cell types, could be affected by the polymorphism we identified.

Our results suggest a link between BDNF-TrkB signaling, hippocampal volumes and antidepressant efficacy in MDD patients. The observation that numerous compounds and procedures, with widely different (and, at times, poorly understood) mechanisms of action, have antidepressant properties [92], has long raised the need for a comprehensive explanatory framework of MDD treatment [15]. One of the leading hypotheses postulates a common effect of antidepressants on brain neuroplasticity [93], modulated at least partially by their effect on BDNF-TrkB signaling [15]. Antidepressant treatment has been shown to increase BDNF in peripheral blood [94], in CSF [95] and in the brain [96]. BDNF in turn enhances neural plasticity in depressed brains, a process thought to be crucial in ameliorating depressive symptomatology [97]; precisely for this reason, it is considered to be a crucial mediator of antidepressant effects [98,99]. In this framework, reduced hippocampal volumes that associate with MDD and poor antidepressant response might reflect impaired neural plasticity processes [14], and antidepressant-induced hippocampal volumes increase, as was repeatedly observed [100,101], might reflect a BDNF/TrkB mediated restoration of neuroplasticity and neurogenesis.

This study has various limitations: the recruitment in a single center and the absence of a healthy control group limits the generalizability of our findings. Our sample size, albeit typical for a neuroimaging study, is smaller than those usually employed in genetic analyses. To account for this, given our research question, we a priori selected polymorphisms of interest instead of employing a genome-wide approach; however, we selected all available SNPs for *BDNF* and *NTRK2* in our genetic array and we applied a strict Bonferroni multiple comparison correction to all our analyses. Likewise, in our MRI analysis, we did not employ a voxel-wise approach, but rather we performed a hypothesis-driven region-of-interest analysis on the hippocampus. Future better-powered multi-center studies could help to overcome these limitations.

Given the real-world nature of our study, patients were prescribed a wide variety of antidepressants drugs, whose effect we did not control for; however, we employed consensus definitions of treatment resistance widely used in real world studies [54]. Furthermore, we could not control for factors known to affect both hippocampal volumes and treatment resistance, such as adiposity and insulin resistance, inflammatory status or childhood trauma. No depression severity cut-off was used as an inclusion criteria; as a result, included patients had varied depression severity levels (Table 1). Albeit, no difference in baseline HDRS scores was identified between TRD and non-TRD patients; whether depression severity influences the relation between *NTRK2* polymorphisms, hippocampal volumes and treatment resistance remains to be determined.

So far, large genome-wide association studies (GWAS) did not find genetic variation in *NTRK2* or *BDNF* regions to associate with antidepressant efficacy [102,103]. Different study populations (both in diagnosis and severity), definitions of treatment resistance and methods employed to assess antidepressant efficacy might all justify this inconsistency. Nevertheless, this implies that our results on treatment resistance must be taken with caution. Likewise, genetic variation in the *NTRK2* region was not found to be associated with hippocampal volumes in GWAS studies, which however were performed mostly on healthy individuals [104,105,106]. Alterations of the BDNF-TrkB signaling pathway might on the other hand be a specific feature of MDD pathophysiology [17], and polymorphisms affecting TrkB expression might contribute to the established reduction in hippocampal volumes in this population [63].

Despite these limitations, this study supports the notion that BDNF/TrkB genetic variations play a role in hippocampal-mediated antidepressant resistance, contributes to a better understanding of MDD pathophysiology and points to possible markers of antidepressant efficacy which, moving towards a personalized approach to treatments, might become increasingly relevant in future real-world clinical practice.

## Figures and Tables

**Figure 1 genes-14-02037-f001:**
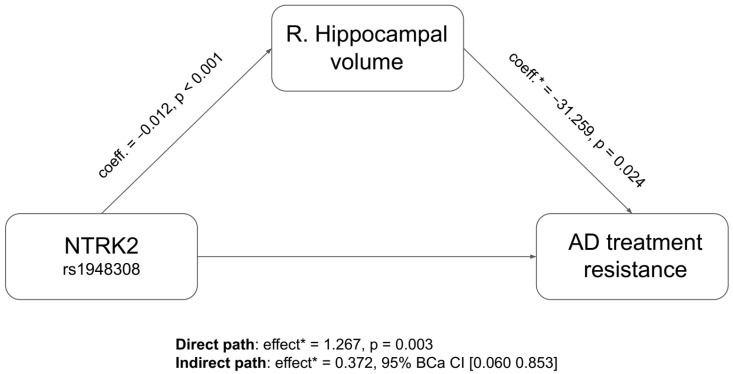
Indirect effect of *NTRK2* polymorphisms on treatment resistance status through right hippocampal volume. AD: antidepressant; BCa CI: Bias-corrected and accelerated bootstrap confidence interval. * log-odds metric.

**Table 1 genes-14-02037-t001:** Clinical and demographic characteristics of the sample.

	Whole Sample (n = 121)	No Treatment Resistance (n = 60)	Treatment Resistance (n = 61)	χ^2^/*t*-Test *p*
Age	50.08 ± 10.26	50.53 ± 9.02	49.64 ± 11.40	0.633
Sex (F/M)	77/44	41/19	36/25	0.287
Education (yrs.)	12.33 ± 3.85	12.06 ± 3.33	12.60 ± 4.32	0.450
Age of onset	32.61 ± 12.48	34.02 ± 12.01	31.15 ± 12.89	0.214
Baseline HDRS	22.02 ± 6.41	22.27 ± 6.32	21.80 ± 6.56	0.694
Duration of hospitalization (days)	27.74 ± 10.40	24.85 ± 7.95	30.79 ± 11.80	0.002
BMI	25.13 ± 4.33	24.77 ± 4.46	25.48 ± 4.20	0.376
R. Hippocampus as % of TIV	0.221 ± 0.017	0.227 ± 0.017	0.216 ± 0.017	<0.001
L. Hippocampus as % of TIV	0.206 ± 0.017	0.211 ± 0.015	0.200 ± 0.016	<0.001

## Data Availability

Data available upon request.

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
