# Peer review of "Association between NTRK2 Polymorphisms, Hippocampal Volumes and Treatment Resistance in Major Depressive Disorder"

_genes, 2023, doi:10.3390/genes14112037_

Round 1
Reviewer 1 Report
Comments and Suggestions for Authors
This study provides valuable insights into genetic factors related to major depression treatment
resistance and their impact on hippocampal volumes.
The paper's structure is generally clear, but minor revisions may be needed for improved clarity
and flow.
• Discuss the adequacy of the sample size and the generalizability of the findings to a
broader population.
• Confirm that the research adhered to ethical guidelines for human research, particularly
regarding blood sampling and data collection.
• The discussion provides valuable insights into the study's findings. However, it could
be strengthened by further exploration of the potential clinical applications and the
limitations of the research. Address potential confounding factors that might influence
hippocampal volumes and treatment response.
• Ensure that the statistical methods and tests employed are thoroughly described and that
any underlying assumptions are clearly stated.
• Can the authors please elaborate on the significance of their findings to patients in
future steps?
• The manuscript should be reviewed for grammar, spelling, and clarity of language
Author Response
This study provides valuable insights into genetic factors related to major depression treatment resistance and their impact on hippocampal volumes.
The paper's structure is generally clear, but minor revisions may be needed for improved clarity and flow.
We thank this Reviewer for all the helpful comments.
- Discuss the adequacy of the sample size and the generalizability of the findings to a broader population.
We discussed these aspects further in the limitation section (page 7).
- Confirm that the research adhered to ethical guidelines for human research, particularly regarding blood sampling and data collection.
We confirmed this aspect in the manuscript (page 3).
- The discussion provides valuable insights into the study's findings. However, it could be strengthened by further exploration of the potential clinical applications and the limitations of the research. Address potential confounding factors that might influence hippocampal volumes and treatment response.
Thanks. We addressed these aspects in the limitations (page 7).
- Ensure that the statistical methods and tests employed are thoroughly described and that any underlying assumptions are clearly stated.
We thank the Reviewer for the suggestion and we checked methods and tests.
- Can the authors please elaborate on the significance of their findings to patients in future steps?
We changed the conclusion section accordingly (page 7 and 8).
- The manuscript should be reviewed for grammar, spelling, and clarity of language
Thanks. We fixed typos and spelling errors.
Reviewer 2 Report
Comments and Suggestions for Authors
In this study the authors present outcomes that suggests a relation between genetic variations in the NTRK2 gene, the hippocampal volume reduction and the resistance of MDD patients to antidepressant treatments. These findings could have implications for our understanding of the mechanisms underlying antidepressant action and MDD pathophysiology.
MAJOR POINTS
The major concern about the quality of this work is related to the methods and results sections. The authors have described clinical and demographic characteristics in detail (Table 1). However, a better and detailed descriptions of the results obtained should be provided in the text in order to provide a fully comprehension of this study that justify the conclusions.
In the methods section, authors described results for 272 comparisons included in Table 1 and 2 (Lines 180 and 187). However, there is no Table 2 in the manuscript or there is something missing in the method description.
The principal results are described in a diagram (Figure 1) that is poorly explained. Which coefficients are describing the relations? The authors should explain it in the text. Also, the p values in the diagram does not corresponds to any test described in the manuscript.
It was impossible to me to access the supplemental material. Please, check this error.
Finally, despite the authors recognize the limitations of this study and prevent about their conclusions, the title is very affirmative and conclusive. It should be changed.
All these issues must be explained before this work can be considered for publication.
MINOR POINTS
Definitions of several acronyms are missing in the text:
Line 50: BDNF
Line 63: ECT
Line 98: SNPs
Line 251: the initial AD is firstly used in the text in line 251 and is not described there but in the figure 1. Please, specify it early in the manuscript (introduction section)
Line 290: GWAS
Line 71: About the expression: “both typical monoaminergic antidepressants” authors should explain to which ones they are referring to
Line 297: check references format of: [87] [88, 89]. Also, the semicolon should be replace by a point and the sentence should be reorganized
Author Response
MAJOR POINTS
- The major concern about the quality of this work is related to the methods and results sections. The authors have described clinical and demographic characteristics in detail (Table 1). However, a better and detailed descriptions of the results obtained should be provided in the text in order to provide a fully comprehension of this study that justify the conclusions.
We thank Reviewer #2 for the constructive comments and important suggestions and we improved the results section accordingly
- In the methods section, authors described results for 272 comparisons included in Table 1 and 2 (Lines 180 and 187). However, there is no Table 2 in the manuscript or there is something missing in the method description.
Polymorphisms distribution and the result of each statistical test were reported in the supplementary materials. The tables referenced in the methods section are supplementary Tables 1 and 2. We re-uploaded supplementary materials and we hope that they are now visible.
- The principal results are described in a diagram (Figure 1) that is poorly explained. Which coefficients are describing the relations? The authors should explain it in the text. Also, the p values in the diagram does not corresponds to any test described in the manuscript.
In the mediation model, direct and indirect effect of NTRK2 on treatment resistance, as well as of HVs on treatment resistance, are expressed in log-odds metric. We specified this aspect in the figure caption and expanded the results section (page 5 and 6).
- It was impossible to me to access the supplemental material. Please, check this error.
We re-uploaded the supplementary materials and we hope that now this problem is fixed.
- Finally, despite the authors recognize the limitations of this study and prevent about their conclusions, the title is very affirmative and conclusive. It should be changed.
We changed the title, accordingly to the Reviewer’s suggestion.
MINOR POINTS
Definitions of several acronyms are missing in the text:
Line 50: BDNF
Line 63: ECT
Line 98: SNPs
Line 251: the initial AD is firstly used in the text in line 251 and is not described there but in the figure 1. Please, specify it early in the manuscript (introduction section)
Line 290: GWAS
Line 71: About the expression: “both typical monoaminergic antidepressants” authors should explain to which ones they are referring to
Line 297: check references format of: [87] [88, 89]. Also, the semicolon should be replace by a point and the sentence should be reorganized
Thanks. We corrected all the above-mentioned issues.
Reviewer 3 Report
Comments and Suggestions for Authors
This cross-sectional study investigated the role of BDNF and NTRK2 polymorphisms in the relationship between hippocampal volume and treatment resistance in MDD. The main finding of the present study is that heterozygotes for the NTRK2 non-coding region (s1948308) had higher chance to treatment resistant than AA or GG homozygotes. This polymorphism has also mediated bilateral hippocampal volumes with treatment resistance. This is an interesting topic which contributes to our understanding of the complex phenomena of TRD.
However, there are some suggestions:
In „Materials and methods “it is claimed that 126 patients were included. However, in table 1 the whole sample included 121 patients.
The authors mentioned that: „We hypothesized that genetic variants of BDNF or NTRK2 genes would associate with hippocampal volume which turn would affect treatment resistance “. However, I cannot see the results for BDNF gene, so please, provide them
Please, mention the treatment that the patients were currently receiving. While TRD patients often receive low-dose antipsychotics (despite not having psychotic symptoms), rTMS; ECT, esketamine, etc., did TRD group differ from non-TRD in respect with treatment?
Given that the baseline HAMD was 22.02 ± 6.41 in the entire sample, and similar in TRD and non-TRD, the patients had moderate depression severity. Please add in „inclusion criteria “if there was a certain HAMD cut-off value.
Please, comment whether the findings may be generalized to patients with more severe depression.
In Discussion, please add more data on the findings of rs1948308 in different conditions and treatment response, and how your data extend the knowledge of this particular polymorphism, and what are the avenues for future studies
I think the limitation is also the lack of healthy control group
Author Response
- In „Materials and methods “it is claimed that 126 patients were included. However, in table 1 the whole sample included 121 patients.
126 patients were originally included in the study; however, after genetic quality check, 5 patients were excluded resulting in a final sample of 121 subjects. This is specified in the methods section, lines 143-144.
- The authors mentioned that: „We hypothesized that genetic variants of BDNF or NTRK2 genes would associate with hippocampal volume which turn would affect treatment resistance “. However, I cannot see the results for BDNF gene, so please, provide them.
No statistically significant results were identified for polymorphisms affecting the BDNF gene region. We specified it in the results section of the manuscript (page 5) and in supplementary Table 2.
- Please, mention the treatment that the patients were currently receiving. While TRD patients often receive low-dose antipsychotics (despite not having psychotic symptoms), rTMS; ECT, esketamine, etc., did TRD group differ from non-TRD in respect with treatment?
Thanks. We specified details on prescribed pharmacotherapies in supplementary Table 3.
- Given that the baseline HAMD was 22.02 ± 6.41 in the entire sample, and similar in TRD and non-TRD, the patients had moderate depression severity. Please add in „inclusion criteria “if there was a certain HAMD cut-off value. Please, comment whether the findings may be generalized to patients with more severe depression.
No specific HAMD cut-off was applied as inclusion criteria. However, all patients received a major depressive disorder with an ongoing depressive episode diagnosis by a staff psychiatrist upon admission. We stressed this aspect in the limitation section (page 7).
- In Discussion, please add more data on the findings of rs1948308 in different conditions and treatment response, and how your data extend the knowledge of this particular polymorphism, and what are the avenues for future studies
We expanded these aspects in the discussion and limitation sections (page 7).
- I think the limitation is also the lack of healthy control group
We agree with the Reviewer and we added this important criticism to the limitation section (page 7).
Reviewer 4 Report
Comments and Suggestions for Authors
Although the article is covering a very interesting topic, the role of BNDF in hippocampal volumes, it is still missing a lot of information. Only one result figure and a small table are presented as the results of this beautiful study. More data analysis of different inputs of the study are still needed. For exmaple, where are the results for the bioinformatic analysis, of that study "using the Infinium PsychArray 24 BeadChip (Illumina, Inc., San 129 Diego), which is a cost-effective, high-density microarray developed in collaboration with the Psychiatric Genomics Consortium and containing 595427 genetic markers associated with psychiatric disorders" ? Where are the results of the brain imaging? If this information is not provided, the study can lose its seriousness.
Comments on the Quality of English LanguageMinor English corrections are needed.
Author Response
- Although the article is covering a very interesting topic, the role of BNDF in hippocampal volumes, it is still missing a lot of information. Only one result figure and a small table are presented as the results of this beautiful study. More data analysis of different inputs of the study are still needed. For exmaple, where are the results for the bioinformatic analysis, of that study "using the Infinium PsychArray 24 BeadChip (Illumina, Inc., San 129 Diego), which is a cost-effective, high-density microarray developed in collaboration with the Psychiatric Genomics Consortium and containing 595427 genetic markers associated with psychiatric disorders" ? Where are the results of the brain imaging? If this information is not provided, the study can lose its seriousness.
We thank Reviewer #4 for the comment. Extended information on genetic data and performed analyses can be found in supplementary materials. We re-uploaded the supplementary file and hope that now it is accessible.
Round 2
Reviewer 2 Report
Comments and Suggestions for Authors
The authors have addressed all of my concerns and suggestions adequately.
I believe that in this form, the manuscript could be considered for publication in the MDPI journal
Author Response
We thank the reviewer for his/her helpful comments
Reviewer 3 Report
Comments and Suggestions for Authors
The authors have accepted all suggestions. I have no further comments, except that please, make sure that all abbreviations in the text are first time mentioned by the full name (i.e., NMDA, GABA, AMPA, TrkB), also in the abstract
Author Response
As suggested by the reviewers we checked that all acronyms are fully spelled the first time they are used